# CLARA: Convex Low-resource Accent-Robust Language Detection in ASR

## Abstract

Globalization and multiculturalism continue to produce increasingly diverse speech varieties. Yet current spoken dialogue systems frequently fail on underrepresented dialects and accents, often misidentifying the input language and causing cascading failures in downstream dialogue tasks. Addressing this dialectal variance under low-resource constraints remains an open challenge, as standard fine-tuning is computationally expensive and prone to overfitting on high-dimensional speech data. We propose Convex Language Detection (CLD), a novel framework that integrates a theoretically grounded convex optimization techniques into the spoken dialogue systems pipeline. Our method is efficiently implemented via multi-GPU Alternating Direction Method of Multipliers (ADMM) in JAX, thus providing global optimality guarantees and fast training in polynomial time. Theoretically, we prove that our convex objective induces certified margin stability and provide rigorous guarantees against feature perturbations. Empirically, we demonstrate sample efficiency and robustness to input dialectical variance, to significantly reduce language misidentification rates for low-resource dialects within high-resource languages. Our open-source package is available at https://anonymous.4open.science/r/CLD-845F/README.md.

## 1 Introduction

Spoken language dialogue systems are now ubiquitous across cultures, countries, and applications. From multimodal agents to everyday voice assistants such as Siri (Apple Inc., 2011), Google Assistant (Google LLC, 2016), and Amazon Echo (Amazon.com, Inc., 2014), conversational user interfaces are becoming increasingly essential in daily life. The critical shared component among these systems is Automatic Speech Recognition (ASR), which transcribes user speech input signals into text for downstream processing by Large Language Models (LLMs). Without accurate transcripts, even the most capable LLMs struggle to infer intent or generate reliable responses. This persistent performance gap between speech input and text input has motivated growing interest in developing more robust ASR systems. For example, recent model families such as Whisper (Radford et al., 2022) demonstrate strong zero-shot generalization across datasets and domains, yet frequently misidentify the input language when confronted with real-world diverse accents and dialectal variation (Kuhn et al., 2024).

This limitation arises in part since existing voice-transcription datasets rarely annotate fine-grained human speech intonations, leading to systematic under-representation of regional dialects even within high-resource languages. Eberhard et al. (2025) notes approximately 380 million people globally speak English as their first language, with the majority of English speakers using English as a second language, over 600 million people are native Hindi speakers, over 1.3 billion people speak various dialects of Chinese, and over 950 million people speak in various Southeast Asian dialects. Notably, although the national language of Singapore is English (arguably the most dominant language in voice datasets), the unique and prevalent dialect of Singaporean accented English has led to the colloquial term "Singlish" (Wee, 2018). The intonation and prosody of Singlish is so distinct that it has been widely studied by linguists: Goh (2016), Hoon (2003), and Rubdy (2007). Despite its prevalence, state-of-the-art ASR models often mistakenly transcribe Singlish in a foreign language (such as Bahasa (Le Page, 1984) and Tamil (Rajan, 2018)). This frequently leads to cascading errors in downstream dialogue or task completion models, and raises significant concerns

about the lack of inclusivity and equitable access to spoken dialogue systems models (Sharma et al., 2022), (Khoury et al., 2025).

Addressing this challenge is technically difficult, since speech patterns vary across age, gender, cultural background, and multilingual experience. Furthermore, spontaneous speech often includes code-switching, disfluencies, and domain-specific vocabulary, all of which further complicate language identification and transcription. These factors create a persistent mismatch between training distributions and real-world applications, even as model capacity continues to scale. Given ongoing global trends toward multicultural and multilingual societies, these failures may disproportionately affect millions of users and raise pressing concerns regarding accessibility, inclusion, and user trust. Recent research has begun to examine dialectal gaps across languages (Kantharuban et al., 2023) and explore improved user experiences in multilingual human–computer interaction (Li et al., 2023).

Despite this momentum, progress in spoken dialogue systems continues to lag behind text-only language modeling due to the comparative scarcity of comprehensive voice data. Audio is significantly more expensive to collect and curate than text, requiring strict quality control, privacy considerations, and access to real human participants. As a result, modern ASR performance is increasingly constrained by a lack of available training data (Beaulieu & Leonelli, 2021), and researchers often utilize an unchanging set of established speech corpora (Serban et al., 2015). These resource constraints help explain why, despite rapid gains in LLM scaling capabilities, robust ASR for diverse real-world speech remains a challenging and important open problem. Therefore a current paradigm is to drive progress in speech models by maximizing the signal extracted from existing, limited and low-resource regimes.

In this paper, we aim to take a step toward democratizing access to spoken dialogue systems that robustly handle user speech input across multicultural backgrounds. We introduce the novel **C**onvex **L**anguage **D**etection (CLD) framework, which leverages theoretically grounded convex optimization techniques for robust language detection under dialectal variation. Our method achieves global optimality in polynomial time, offers improved sample efficiency with stronger generalization guarantees. This efficiency is essential for end-to-end spoken dialogue systems, which must maintain sub-500ms latency to preserve natural human conversational timing (Meyer, 2023). We further optimize for fast training and inference by implementing our method in JAX (Bradbury et al., 2021) and solving the foundational convex program using Alternating Direction Method of Multipliers (ADMM) techniques (Boyd et al., 2011). To the best of our knowledge, this represents the first practical application of convex optimization reformulations on spoken dialogue systems for robust language classification.

Our contributions are as follows:

- We propose **C**onvex **L**anguage **D**etection (CLD), a fast sample-efficient algorithm for robust spoken language classification among low-resource language regimes. We demonstrate CLD's strong efficiency in the critical and challenging dialect-identification task in ASR models. Section 3 formally introduces the CLD algorithm and methodology.

- We recast the CLD network as a convex program and prove certified robustness. By characterizing the variation norm induced by our convex program, we derive exact logit-Lipschitz constants and prove certified margin stability against hidden-feature perturbations (Theorem 1). This provides a computable, data-dependent certificate of label invariance, ensuring that the model's predictions remain stable within a guaranteed radius in Section 4.

- We validate the practical efficiency of CLD with improved accuracy, reduced training time, and significantly lower memory requirements. We benchmark across three ASR models, two datasets, and conduct both binary and multiclass language classification tasks for low-resource dialects within high-resource languages. Our experiments span five languages and twenty-four sub-dialects. Results are presented in Section 5.

- Our `pip installable` open-source multi-GPU JAX package[1] is provided for ease of reproducibility and continued work.

---

[1] https://anonymous.4open.science/r/CLD-845F/README.md

## 2 RELATED WORK

**Performance on Multilingual Tasks.** Foundational multilingual ASR models like Whisper have up to 1.55 billion parameters and have been trained on more than 99 languages (Radford et al., 2022). However the vast majority of these models perform the best on English, with performance dropping significantly on lower resource languages due to lack of training data (Graham & Roll, 2024). This has recently encouraged much work in the field of improving low-resource ASR performance. For example, the authors Bansal et al. (2018), Khare et al. (2021), and Stoian et al. (2020) propose using transfer learning via pretraining techniques to improve cross-lingual transfer. This requires expansive amounts of speech data in existing high-resource languages but with text transliterated to the target low-resource language. Essentially the mapping serves to encourage increased sharing between the output spaces of both languages, yet the success of pretraining is not well defined. The high-resource and low-resource language must share a certain amount of unclear "basis similarity" in linguistics for this to be feasible. During the course of pretraining on extremely large datasets, the powerful base ASR model also experiences catastrophic forgetting, leading to overall deterioration in performance.

**Improving Language Detection.** Even within high resource languages such as English and Mandarin, there exist many distinct dialects which state-of-the-art ASR models struggle to identify correctly. The recent works of Li et al. (2024), Weninger et al. (2019), and Wang et al. (2025) aim to implement prosody-assisted speech systems, or bidirectional Long-Short-Term Memory networks to better model acoustic context. With the the rise in popularity of spoken dialogue models, other researchers (Reitmaier et al., 2022) have focused on more clearly identifying the challenges ASR models face with low-resource languages. These methods share the common weakness of being heavily dependent on large fine-tuning datasets with a learning rate that is typically ten times smaller than standard supervised fine-tuning learning rates (Wilson & Martinez, 2001), (Liu et al., 2024), (de Zuazo et al., 2025).

**Certified Robustness in Deep Learning.** Ensuring model reliability under input perturbations is a critical area of study, which is typically addressed via empirical defenses like adversarial training (Madry et al., 2018). However, empirical defenses typically do not provide formal guarantees. This has driven interest in certified robustness and information theory, which aims to provide provable bounds on classifier consistency (e.g., via randomized smoothing or Lipschitz analysis) (Cohen et al., 2019; Tsuzuku et al., 2018). In the audio domain, robustness is usually evaluated against environmental noise rather than dialectal shifts. By leveraging the specific geometry of our convex reformulation, we derive a variation-norm certificate that bounds the Lipschitz constant of the detection head directly. Unlike black-box certification methods, our approach yields a constructive certificate of margin stability derived explicitly from the optimization objective.

Additional discussion on related work regarding convex reformulated models and the need for sample efficiency is presented in Appendix B.

## 3 METHODOLOGY

The **C**onvex **L**anguage **D**etection (CLD) methodology is formally presented: Section 3.1 provides preliminary background on the convex reformulated program of two-layer ReLU networks, and Section 3.2 present its integration with ASR model architecture to yield the CLD framework.

### 3.1 CONVEX REFORMULATION OF TWO-LAYER RELU NETWORKS

**Background.** We observe the standard two-layer ReLU network as $f(x) = \sum_{j=1}^{m} (\Theta_{1j} x)_+ \theta_{2j}$. Here $x \in \mathbb{R}^d$ denotes the input, $\Theta_1 \in \mathbb{R}^{m \times d}$ and $\theta_2 \in \mathbb{R}^m$ are the layer weights, and $(\cdot)_+ = \max\{\cdot, 0\}$ is the ReLU activation. Given training labels $y \in \mathbb{R}^n$, the model's standard non-convex training objective can be seen as

$$\min_{\Theta_1, \theta_2} \ell\big(f_{\Theta_1, \theta_2}(X), y\big) + \frac{\beta}{2} \sum_{j=1}^{m} \left( \|\Theta_{1j}\|_2^2 + (\theta_{2j})^2 \right), \tag{1}$$

with loss function $\ell : \mathbb{R}^n \to \mathbb{R}$, data matrix $X \in \mathbb{R}^{n \times d}$, and regularization $\beta \geq 0$. Equation 1 presents a challenging non-convex optimization problem, and its minimization is sensitive to

hyperparameter tuning (i.e. learning-rate selection). These issues become significantly amplified in large-scale speech applications, where models are more expensive to train than their text-input counterparts. High-dimensional audio data also practically prohibits comprehensive grid-search of all hyperparameters (Sainath et al., 2013). Our goal is to retain the expressiveness of equation 1 while employing the stability and reliability of classic convex optimization.

**Convex reformulation.** Pilanci & Ergen (2020) show that equation 1 admits an equivalent convex neural network (cvxNN) representation when the hidden width satisfies $m \geq m^*$ for some $m \leq n + 1$. The reformulation relies on characterizing all possible ReLU activation patterns induced by $X$. Each pattern corresponds to a diagonal matrix selecting rows of $X$, and the full activation-pattern set is $\mathcal{D}_X = \{D = \mathrm{diag}(\mathbf{1}(Xv \geq 0)) : v \in \mathbb{R}^d\}$. The cardinality satisfies $|\mathcal{D}_X| = \mathcal{O}(r(n/r)^r)$ with $r = \mathrm{rank}(X)$ (Pilanci & Ergen, 2020). Since $|\mathcal{D}_X|$ can grow exponentially, enumerating all patterns is infeasible. Instead, we sample $P$ patterns $\{D_i\}_{i=1}^P$ and solve the equivalent convex optimization problem

$$\min_{(v_i, w_i)_{i=1}^P} \ell\left(\sum_{i=1}^P D_i X(v_i - w_i),\, y\right) + \beta \sum_{i=1}^P (\|v_i\|_2 + \|w_i\|_2) \tag{2}$$

$$\text{s.t. } v_i, w_i \in \mathcal{K}_i, \qquad \forall i \in [P].$$

Under mild conditions, equation 2 retains the same optimal solution as the non-convex formulation of equation 1 (Mishkin et al., 2022). Even in cases where the solutions differ marginally, the results of Kim & Pilanci (2024) show that the discrepancy is negligible in practice. The recent introduction of CRONOS (Feng et al., 2024) also demonstrated successful application of the convex reformulation on deep learning LLM classification tasks with GPT-2 (Radford et al., 2019). Therefore we can confidently leverage this principled convex surrogate in our study to both preserve the expressive capacity of our models, and enable stable optimization by eliminating dependence on brittle hyperparameters.

### 3.2 CONVEX LANGUAGE DETECTION ALGORITHM

**Training for Scale.** In order to tractability work with audio input data our three main algorithmic desiderata are – scale, speed, and robustness. Scale is a particularly crucial since we experiment with high-dimensional and sensitive multi-class speech data. To address this, we first extract *hidden representations* from the ASR encoder, then solve the resulting problem using a multi-GPU, batched implementation of CRONOS in JAX for enhanced parallelization. This yields our CLD framework: a lightweight detection head that operates directly on encoder features to identify the input language before decoding. Formally, given an input waveform $x$ sampled from dataset $(x_i, y_i)_{i=1}^N$, the encoder produces a representation $h$ from which CLD predicts a language token $\hat{y}$. The decoder then generates the transcript $\hat{t}$ conditioned on this token. Algorithm 1 outlines the offline CLD training procedure.

---

**Algorithm 1** CLD Training (Offline)

---

**Input:** Whisper Encoder $\mathcal{E}$, Dataset $\mathcal{D}_{\text{train}}$, parameters $\rho, \beta$

**Output:** Trained Convex Detection Head $\hat{f}_{\text{cvx}}$

1 **for** $i \leftarrow 1$ **to** $N$ **do**

2 $\quad h_i \leftarrow \mathcal{E}(x_i)$          ▷ Extract hidden states for sample $i$

3 Initialize ADMM variables $(\mathbf{v}, \mathbf{w}, \mathbf{u})$.

   **repeat**

4 $\quad (\mathbf{v}, \mathbf{w}) \leftarrow \arg\min_{\mathbf{v}, \mathbf{w}} \left[ \ell\left(\sum_{p=1}^P D_p H(\mathbf{v}_p - \mathbf{w}_p), y\right) + \beta \sum_{p=1}^P (\|\mathbf{v}_p\|_2 + \|\mathbf{w}_p\|_2) + \frac{\rho}{2}\| \cdot \|_2^2 \right]$

                                                 ▷ Primal Update: Minimize Augmented Lagrangian

   $\mathbf{u} \leftarrow \mathbf{u} + \rho \cdot (\mathbf{v} - \mathbf{w})$          ▷ Dual Update: Update Lagrange Multiplier

5 **until** *convergence*

6 **return** *Store trained weights as* $\hat{f}_{\text{cvx}}$

---

**Low Latency Inference.** Algorithm 2 summarizes the CLD online inference pipeline. Fast online inference in achieved by augmenting the Whisper encoder–decoder pipeline with the trained convex module. Given an input audio waveform $x$, the encoder $\mathcal{E}$ first produces a hidden representation $h$, which is passed through the trained convex head $\hat{f}_{\text{cvx}}$ to predict the language token $\hat{y}$. This prediction is computed in a single lightweight forward pass, ensuring sub-500ms latency. The predicted token $\hat{y}$ is then supplied to the Whisper decoder $\mathcal{D}$ as the initialization token, enabling the decoder to accurately generate the final transcription $\hat{t}$ conditioned on the detected language. Thus, CLD integrates seamlessly into the ASR pipeline and improves transcription robustness while incurring negligible latency at inference.

---

**Algorithm 2** CLD Inference (Online)

---

**Input:** Whisper Encoder $\mathcal{E}$, Decoder $\mathcal{D}$, Trained Head $\hat{f}_{\text{cvx}}$, Input Audio $x$
**Output:** Predicted Language $\hat{y}$, Transcription $\hat{t}$
7  $h \leftarrow \mathcal{E}(x)$                                                      ▷ Encoder forward pass to get hidden state
8  $\hat{y} \leftarrow \arg\max \hat{f}_{\text{cvx}}(h)$                              ▷ Convex head predicts language label
9  $\hat{t} \leftarrow \mathcal{D}(x\,;\,\text{init\_token} = \hat{y})$              ▷ Decoder uses $\hat{y}$ to initialize generation
10  **return** $(\hat{y}, \hat{t})$

---

## 4  THEORETICAL ANALYSIS

In this section we show that the CLD detection head trained via the convex program in Eq. 2 is Lipschitz-stable in the encoder feature space and that the convex program induces a computable robustness certificate. This validates that bounded perturbations of the encoder output $E(x)$ cause at most linear degradation of the one-vs-rest margin, so any example with sufficiently large initial margin enjoys a certified radius of label invariance. Complete proofs and supporting lemmas are provided in Appendix A.

### 4.1  MARGIN STABILITY IN HIDDEN FEATURES

We formally define the CLD head as a multi-class classifier on encoder features and quantify how perturbations in those features affect its predictions. Let $E : \mathbb{R}^T \to \mathbb{R}^d$ denote the ASR encoder, and $h = E(x) \in \mathbb{R}^d$ be the hidden features. The CLD detection module $f : \mathbb{R}^d \to \mathbb{R}^K$ is trained by the convex program in Eq. 2. By the cvxNN construction (Section 3.1), the optimal detection head admits a finite two-layer ReLU representation with the same objective value as the convex program up to negligible approximation from activation-pattern sampling. We now introduce the one-vs-rest classification margin and the variation norm, and use them to derive Lipschitz and margin-stability guarantees for $f$.

**Definition 1** (One-vs-Rest classification margin). For $y \in \{1, \ldots, K\}$ and logits $f(h) = (f_1(h), \ldots, f_K(h))$, define the classification margin as

$$\text{mar}(h, y) := f_y(h) - \max_{k \neq y} f_k(h).$$

**Definition 2** (Variation norm). The variation norm of $f$ is defined as

$$\|f\|_{\text{var}} := \inf \left\{ \sum_{j=1}^m \|a_j\|_2 \|u_j\|_2 \;:\; f(h) = \sum_{j=1}^m a_j [u_j^\top h]_+ \right\}.$$

*Lemma* 1 (Logit Lipschitzness). If $f$ admits a representation of equation 2, then for any $h, h' \in \mathbb{R}^d$,

$$\|f(h) - f(h')\|_\infty \leq \|f\|_{\text{var}} \|h - h'\|_2.$$

**Theorem 1** (Margin stability under hidden-feature perturbations). *Let $f$ be the detection head given by Eq. 2. For any $y \in \{1, \ldots, K\}$ and any $\delta \in \mathbb{R}^d$,*

$$\text{mar}(h + \delta, y) \geq \text{mar}(h, y) - 2\|f\|_{\text{var}} \|\delta\|_2. \tag{3}$$

*Consequently, if $\|\delta\|_2 < \text{mar}(h, y)/(2\|f\|_{\text{var}})$, the predicted class is unchanged.*

Furthermore, if the encoder $E$ is $L_E$-Lipschitz, i.e., $\|E(x) - E(x')\|_2 \le L_E\|x - x'\|_2$, then

$$\mathrm{mar}(E(x + \eta), y) \ge \mathrm{mar}(E(x), y) - 2\,\|f\|_{\mathrm{var}}\,L_E\,\|\eta\|_2,$$

and the predicted class is preserved whenever $\|\eta\|_2 < \mathrm{mar}(E(x), y)/(2\|f\|_{\mathrm{var}}L_E)$. Proof of Theorem 1 is presented in Appendix A.1.

## 4.2 Robustness Certificates from the Convex Program

The previous subsection shows that the variation norm $\|f\|_{\mathrm{var}}$ controls both the logit Lipschitz constant and the stability of the classification margin under hidden-feature perturbations. We now relate this quantity to the solution of the convex program in Eq. 2. We specifically show that any feasible convex solution $(v_p, w_p)_{p=1}^P$ yields an explicit upper bound on $\|f\|_{\mathrm{var}}$, and that this bound can be expressed in terms of block norms (or Frobenius norms) of the optimization variables. This gives a practical, data-dependent robustness certificate: after training CLD, we can read off a certified Lipschitz constant and margin radius directly from the learned weights.

**Proposition 1** (Variation-norm certificate from the convex penalty). *Let $\{(v_p, w_p)\}_{p=1}^P$ denote the variables of Eq. 2. Then*

$$\|f\|_{\mathrm{var}} \le \mathcal{B}_{\mathrm{cvx}} \quad \text{with} \quad \mathcal{B}_{\mathrm{cvx}} := \sum_{p=1}^P \big(\|v_p\|_2 + \|w_p\|_2\big),$$

*interpreting $\|\cdot\|_2$ blockwise (e.g., columnwise $\ell_2$ with a sum across classes). Consequently,*

$$\mathrm{mar}(h + \delta, y) \ge \mathrm{mar}(h, y) - 2\,\mathcal{B}_{\mathrm{cvx}}\,\|\delta\|_2.$$

*If the non-convex two-layer form with penalty $\frac{\beta}{2}\sum_j(\|u_j\|_2^2 + \|a_j\|_2^2)$ is used instead, then by AM–GM (Appendix A.5), $\|f\|_{\mathrm{var}} \le \frac{1}{2}\sum_j(\|u_j\|_2^2 + \|a_j\|_2^2)$, so larger $\beta$ tightens the certified radius.*

In addition, if the logits share blocks (group-sparse outputs), one obtains

$$\|f(h) - f(h')\|_\infty \le \sum_g w_g\|A_g\|_2\|U_g\|_2\,\|h - h'\|_2,$$

hence the margin bound with $\|f\|_{\mathrm{var}}$ can be replaced by the weighted group sum.

**Proposition 2** (Training-set representation). *Let $\{(v_i, w_i)\}$ be any feasible point of equation 5. Then the training predictions equal those of a (vector-valued) two-layer ReLU network with at most $2PK$ hidden units:*

$$f(H) = \sum_{i=1}^P \sum_{k=1}^K \Big(e_k\,[Hv_{i,k}]_+ - e_k\,[Hw_{i,k}]_+\Big),$$

*where $e_k$ are the standard basis vectors in $\mathbb{R}^K$. Equivalently, this network has hidden weights $\{u_{i,k}^+, u_{i,k}^-\} = \{v_{i,k}, w_{i,k}\}$ and output weights $\{a_{i,k}^+, a_{i,k}^-\} = \{e_k, -e_k\}$.*

**Theorem 2** (cvxNN $\Rightarrow$ variation-norm certificate). *Let $f$ be represented as in Proposition 2. Then*

$$\|f\|_{\mathrm{var}} \le \widehat{\mathcal{B}}_{\mathrm{cvx}}^{(2,1)} := \sum_{i=1}^P \big(\|v_i\|_{2,1} + \|w_i\|_{2,1}\big).$$

*If equation 5 uses Frobenius penalties instead, then*

$$\|f\|_{\mathrm{var}} \le \sqrt{K}\,\widehat{\mathcal{B}}_{\mathrm{cvx}}^{\mathrm{F}} \quad \text{with} \quad \widehat{\mathcal{B}}_{\mathrm{cvx}}^{\mathrm{F}} := \sum_{i=1}^P \big(\|v_i\|_F + \|w_i\|_F\big).$$

Proof of Theorem 2 is provided in Appendix A.4.

## 5 EXPERIMENTS

In this section we present main experimental results and discussion. Section 5.1 provides details on our dialect datasets for both binary and multiclass experiments. Section 5.2 presents detailed evaluation of performance metrics. We report on seminal Word Error Rate (WER) metrics, as well as CER, language detection accuracy, wall-to-wall training time and computational efficiency metrics. Our baseline models include: Whisper-small, Whisper-large-v3, and MMS-1b (Pratap et al., 2023).

We additionally benchmark CLD against the ASR's default language detection as well as a traditional lightweight neural network (NN) model commonly used for language identification in ASR. The NN consists of a linear projection from the encoder dimension to a 256-dimensional hidden layer, followed by a ReLU activation, dropout for regularization, and a final linear layer mapping to the output classes.

### 5.1 SPEECH DATASETS

We curate a dataset of multilingual voice transcriptions across high-resource languages and their low-resource sub-dialects. As a primary source of transcription data, we used the Common Voice (v23) Dataset (Ardila et al., 2019). We additionally supplement this with several other custom-sourced dialect datasets to supplement regional speech variance. For one, we selected the Singaporean-English dialect, which has previously shown high error rates during voice transcription (Fong et al., 2002). Through the Info-communications and Media Development Authority (Info-comm Media Development Authority (IMDA), 2025) of Singapore, we were given direct access to the National Speech Corpus (NCS): the first Singapore English corpus. We also use the Lahaja dataset (Javed et al., 2024), a benchmark comprising 12.5 hours of Hindi speech from 132 speakers across 83 Indian districts, for regional Hindi dialects. We normalize and augment all audio files via the following techniques: time stretching, volume gain, pitch shift, and augmented background noise (via MUSAN (Snyder et al., 2015)), which are all used to simulate real-world variability and improve robustness. Our experiments encompass two categories:

**Binary Classification.** We select English and Mandarin as the highest-resource languages in existing seminal datasets, which still display some of the lowest accuracy in language prediction for accented speech. This is largely due to the high variance of dialects and accents present (Weninger et al., 2019) in these two languages. For example, Whisper-small strongly achieves 100% accuracy on Midwestern English, drops to 91.8% accuracy on Wales accented English, yet only achieves 61.4% accuracy on Malaysian accented English (Table 6). We select 5 regional dialects per language and perform ablation studies on training sample sizes spanning from 100 to 10 000 samples per language. This quantitatively establishes CLD's performance in low-resource languages.

**Multiclass Classification.** For the multiclass classification task, we select a total of 5 languages: English, Chinese, Indonesian, Malaysian, Hindi. This selection was made to establish a challenging classification boundary, as these languages share linguistic and geographical proximity. Such regional influences often cause misidentification—for instance, Singaporean English is frequently confused with Malay or Indonesian. To maintain a low-resource experimental setting, we curated a total of 16 000 training samples spanning these 5 languages and incorporating 24 unique accents. This resulted in approximately 3200 samples per language and 666 samples per accent.

### 5.2 RESULTS AND DISCUSSION

**Binary Classification in Low-Resource Regimes.** We validate the performance of CLD augmented Whisper-small in the low-resource setting. Ablation studies on training sample sizes of: 100, 500, 1000, 10 000 [2] is conducted. Figure 1 reports seminal WER and language detection accuracies as sample size scales. Both the vanilla NN and the fine-tuned Whisper-Small (WSP-SFT) exhibit a clear correlation where lower Word Error Rate (WER) corresponds with higher detection accuracy as the sample size increases. This empirically validates the intuition that traditional neural networks require large volumes of data for optimal performance—a challenging prerequisite when working with low-resource dialects where data is inherently scarce. Conversely, our novel CLD model demonstrates visibly consistent performance across all sample sizes, achieving a minimum

---

[2]We ensure fair comparison and utilize the same 1844 sample size test dataset for evaluation.

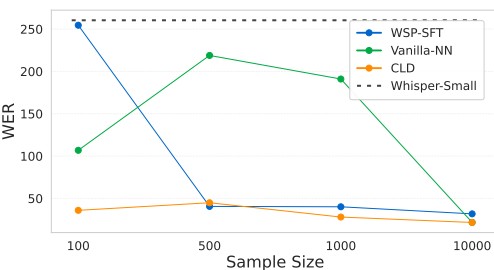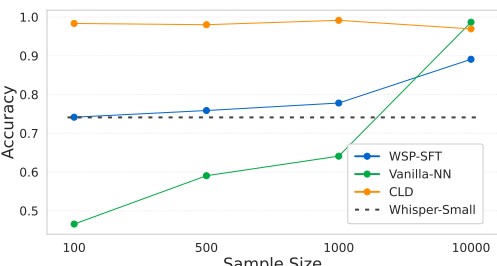

Figure 1: WER (left) – lower is better. Accuracy (right) – higher is better. CLD shows robustness and competitive performance regardless of sample size. Competing methods require more data for better performance.

detection accuracy of 96.94% with 10 000 samples and a maximum of 99.14% with 1000 samples. This confirms its high sample efficiency and strong resilience in low-resource settings. CLD also achieves the lowest WER of 21.62 among all models at the 10 000 sample size, demonstrating that the CLD model is ideally positioned for challenging low-resource regimes. Appendix C provides additional Character Error Rate (CER) plots.

| Model | Training Time (s) | Compute Cost (TFLOPs) |
|---|---|---|
| WSP-SFT | 1,096.74 | 239,528 |
| vanilla-NN | 840.30 | 183,521 |
| **CLD (ours)** | **64.45** | **14,075** |

Table 1: Training efficiency of fine-tuned Whisper-Small (WSP-SFT), vanilla-NN, and CLD. Our method demonstrates substantive compute and training time efficiency.

**CLD is Fast and Efficient.** Table 1 shows that the CLD model achieves a training time of just $64.45$ seconds—approximately 7.7% of the runtime of a standard vanilla NN, while requiring 13x fewer TFLOPs. This efficiency derives from the convex reformulation solved via ADMM (Appendix F) and implemented in JAX, which enables highly parallelizable updates and rapid convergence. Unlike the vanilla NN which requires multiple passes and steps across the dataset for convergence with necessary hyperparameter grid search, the convex program uses a unique global optimum. This allows us to solve directly to the global minima. Together, these properties establish CLD as both fast and efficient, offering a more practical alternative to conventional largely heuristic driven performance among traditional architectures.

**Dialect Variation and Performance.** Table 3 summarizes the performance across dialects at the 500 sample size. Additional numerical results as presented in Tables 2 - 4 of Appendix 5.2. A traditional neural network (NN) can achieve 100% classification accuracy on a subset of English accents, which is expected given the imbalance of pretraining data and the dominance of English representations in the Whisper feature space. However, this also induces a strong classification bias: the model defaults to predicting English, leading to 88.91% of Chinese samples being misclassified as English. In particular, the NN attains achieves 2.72% accuracy on Pu-Xian Chinese, highlighting the English-centric behavior of models trained on imbalanced datasets and the resulting degradation on low-resource dialects.

In contrast, our CLD framework achieves **uniformly high accuracy** across all accents in both languages. Despite Min Dong Chinese having only 71 training samples, CLD still reaches 88.73% accuracy (the fine-tuned NN model only achieves 29.58% accuracy), and exceeds 94% accuracy on all remaining dialects. These results demonstrate the robustness and low variance of CLD even in highly low-resource settings, validating its effectiveness for diverse dialectal speech.

**Low-Resource Dialects in High-Resource Languages.** Although large-scale audio–text datasets have driven recent advances in ASR, high-quality speech data remains costly to collect and difficult to annotate, particularly for languages with substantial dialectal variation. Even the seminal corpora underlying modern ASR models (e.g., the reported 680 000+ hours used to train Whisper by

Radford et al. (2022)) concentrate heavily on a small set of dominant high-resource speech within English and Mandarin, leaving many regional dialects and accented speech varieties underrepresented. This imbalance yields a critical performance gap: speakers of dialects such as Singaporean English ("Singlish") or regional Mandarin varieties experience significantly degraded transcription quality. For example, Table 6 shows Whisper achieving 100% accuracy on Midwestern English but only 61.4% on Malaysian-accented English, despite both being the same language. These disparities highlight that even widely adopted ASR models frequently misidentify dialectal speech, limiting accessibility, reliability and trust for millions of users.

| Language | Accent / Dialect | Size | Correctly Predicted Samples | | | | Accuracy | | | |
|---|---|---|---|---|---|---|---|---|---|---|
| | | | WSP | WSP-SFT | NN | CLD | WSP | WSP-SFT | NN | CLD |
| English | Hindi-accented | 190 | 176 | 177 | 190 | 186 | 0.9263 | 0.9316 | **1.0000** | 0.9789 |
| English | Malaysian-accented | 215 | 136 | 124 | 215 | 214 | 0.6326 | 0.5767 | **1.0000** | 0.9953 |
| English | Singaporean-accented | 205 | 166 | 162 | 205 | 205 | 0.8098 | 0.7902 | **1.0000** | **1.0000** |
| English | Pakistani-accented | 189 | 182 | 182 | 189 | 187 | 0.9630 | 0.9630 | **1.0000** | 0.9894 |
| English | American | 204 | 195 | 194 | 204 | 203 | 0.9559 | 0.9510 | **1.0000** | 0.9951 |
| Chinese | Min Dong / Fuzhou | 71 | 7 | 15 | 21 | 63 | 0.0986 | 0.2113 | 0.2958 | **0.8873** |
| Chinese | Pu-Xian | 216 | 32 | 56 | 44 | 208 | 0.1481 | 0.2593 | 0.2037 | **0.9630** |
| Chinese | Hong Kong Cantonese | 184 | 121 | 132 | 5 | 174 | 0.6576 | 0.7174 | 0.0272 | **0.9457** |
| Chinese | Taiwanese Mandarin | 181 | 176 | 179 | 12 | 181 | 0.9724 | 0.9890 | 0.0663 | **1.0000** |
| Chinese | Mainland Mandarin | 205 | 187 | 190 | 13 | 202 | 0.9122 | 0.9268 | 0.0634 | **0.9854** |
| **Total** | | 1860 | 1378 | 1411 | 1088 | 1823 | 0.7077 | 0.7207 | 0.5576 | **0.9695** |

Table 3: Multi-dialect classification accuracy between English and Chinese across 10 accents for (Whisper-baseline) WSP, WSP-SFT, NN, and CLD at 500 samples.

| Language Classifier | Detection Accuracy | | | WER | | | CER | | |
|---|---|---|---|---|---|---|---|---|---|
| | WSP | WSP-L | MMS-1B | WSP | WSP-L | MMS-1B | WSP | WSP-L | MMS-1B |
| Default | 0.7154 | 0.8033 | 0.6701 | 139.37 | 40.4026 | 51.88 | 73.85 | 21.80 | 27.61 |
| NN | 0.7581 | 0.9695 | 0.8602 | 58.58 | 28.83 | 48.28 | 37.13 | 15.66 | 23.66 |
| **CLD (ours)** | **0.9715** | **0.9806** | **0.9702** | **31.74** | **28.60** | **45.27** | **17.84** | **15.37** | **21.58** |

Table 5: Word Error Rate (WER), Character Error Rate (CER), and Detection Accuracy metrics between Whisper-small (WSP), Whisper-large-v3 (WSP-L), and MMS-1B using Default, NN, and CLD language classifiers.

**Multiclass Classification in Low-Resource Regimes.** Table 5 shows reports seminal metrics on the up-scaled multiclass experiment for Whisper-small, Whisper-large-v3, and MMS-1B across 5 languages. The standard NN struggles to scale to multiple classes (with the exception of Whisper-large-v3 at potential higher training cost), while our novel CLD method wins in all evaluation metrics. CLD maintains strong generalization, achieving as high as a 44.78% increase in detection accuracy and a 12.74% decrease in WER compared to the baseline. For fair comparison we note that larger models like Whisper-large-v3 and MMS-1B in fact have more accurate language detectors as a baseline. This further motivates that augmenting these pre-trained models with the modular CLD architecture can increase accuracy and decrease WER by up to 29.21%. These results demonstrate the scaling potential of the CLD model in multiclass settings to improve the performance of pretrained larger models.

Figure 2 presents confusion matrices for our CLD network and vanilla NN on Whisper-small. While the vanilla NN performs well on Hindi (hi) and Indonesian (id) with accuracies of 93% and 98%, its performance drops sharply for other languages, reaching only 44% on Chinese (zh). It also exhibits systematic confusions: misclassifying Chinese (zh) as Indonesian (id) in 34% of cases and Malay (ms) as Indonesian (id) in 23%. In contrast, our novel CLD model shows strong diagonal dominance across all languages, with its **lowest accuracy still at 95%** in en, counteracting the English-centric bias noted in Section 5.2. These results highlight the stability and consistency of the novel CLD model in multiclass language detection, particularly in diverse real-world conditions.

**Real Human Evaluation.** We validate our results with real human feedback. The tests were conducted on five humans in Singapore speaking English (EN) and ten humans in South-East Peoples Republic of China speaking Mandarin (ZH). Additional numerical tables of results and performance plots are presented in Appendix 7. This is since most metrics are ultimately a surrogate for real

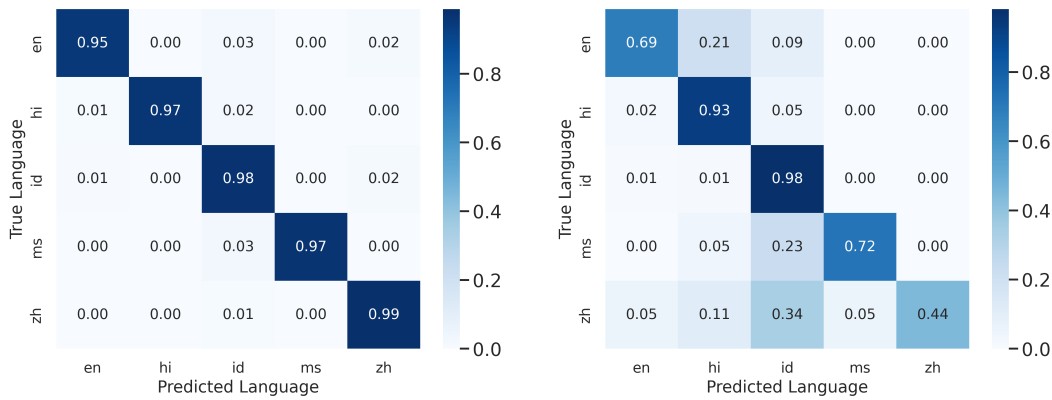

Figure 2: Language Classification Performance between CLD (left) and Vanilla NN (right) with Whisper-small across English (en), Chinese (zh), Indonesian (id), Malaysian (ms), Hindi (hi).

human evaluation, and often fail to capture real human intent. Human testers were instructed to assume the position of a general guest in a hospitality setting requesting an item. This ensures a precise and consistent conversational domain across all models with ease of evaluation. One example of Whisper's output is below:

> **Concierge:** Hello Mr. Kevin Fong, this is Lucy at the front desk. How may I help you?
> **Guest:** Baru keadaan seperti seorang seorang seorang seperti seorang, seorang seorang berada di dalamnya.
> **Concierge:** I apologize, we'll send someone up right away. Do you need anything else?
> **Guest:** No, thank you.

Notably, although the human "guest" was a local Singaporean person speaking naturally in his native English, the Whisper model detected and transcribed this incorrectly into Bahasa. Experiments with a larger traditional NN model for language detection increased performance accuracy marginally, yet when evaluated for Singaporean English frequently mis-transcribed Mandarin characters instead (and vice versa). We discovered a new type of error also arose from the vanilla NN detection head: local accents and dialects introduced errors such as the user speaking *'Both hot and cold settings'* to *'Both hood and coat setting'*. In contrast, our novel CLD framework achieved the fastest inference and most accurate results demonstrating robustness to input dialect variance. Achieving both minimal word errors and the smallest number of wrong language detections.

## 6 CONCLUSION

We introduced the Convex Language Detection (CLD) framework, providing a lightweight, fast, and theoretically grounded approach for robust language identification in Automated Speech Recognition (ASR). By reframing the classification problem as a convex optimization program, we achieved certified robustness and superior sample efficiency, allowing CLD to maintain consistent, high accuracy regardless of the training data size. CLD effectively mitigated the English-centric failure modes of existing ASR pipelines, demonstrating its effectiveness in challenging low-resource and diverse dialectal speech conditions. While highly efficient, a current constraint is that the convex head is applied to fixed encoder features; future work will explore integrating CLD more deeply within the traditional ASR model to enable end-to-end continuous representation learning, and investigate applying this convex-analytic framework to other components within multimodal agentic models.

ACKNOWLEDGMENTS

During initial submission, we respectively omit the acknowledgments section in order to adhere to the the double-blind reviewing process.

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

# A PROOF OF MAIN RESULTS AND CERTIFICATES FOR MARGIN STABILITY

This sections links the convex program in Eq. (3) and the two-layer ReLU representation used in Theorem 1, together with computable certificates that translate directly into certified radii.

## A.1 PROOF OF LEMMA 1

*Proof.* Write $f_k(h) = \sum_j a_{j,k} [u_j^\top h]_+$. Since $t \mapsto [t]_+$ is 1-Lipschitz, $|f_k(h) - f_k(h')| \leq \sum_j |a_{j,k}| |u_j^\top (h - h')| \leq \sum_j \|a_j\|_2 \|u_j\|_2 \|h - h'\|_2$. Maximizing over $k$ and taking the infimum over representations yields the claim. $\square$

**Proof of Theorem 1**

*Proof.* Let $k^\star(h) = \arg\max_{k \neq y} f_k(h)$. Then

$$\mathrm{mar}(h + \delta, y) - \mathrm{mar}(h, y) = \big(f_y(h + \delta) - f_y(h)\big) - \big(\max_{k \neq y} f_k(h + \delta) - f_{k^\star(h)}(h)\big).$$

Each difference is $\leq \|f\|_{\mathrm{var}} \|\delta\|_2$ by Lemma 1, yielding equation 3. $\square$

## A.2 ACTIVATION PATTERNS AND PATTERN CONES

For $u \in \mathbb{R}^d$ and a data matrix $H \in \mathbb{R}^{n \times d}$, define the training activation pattern

$$D(H, u) = \mathrm{diag}\big(\mathbf{1}\{Hu \geq 0\}\big) \in \{0, 1\}^{n \times n}.$$

Given a fixed pattern $D \in \{0, 1\}^{n \times n}$, define its associated *pattern cone*

$$\mathcal{K}(D) := \big\{ v \in \mathbb{R}^d : (2D - I) Hv \geq 0 \text{ (entrywise)} \big\}.$$

Then $v \in \mathcal{K}(D)$ if and only if $D(H, v) = D$ (up to measure-zero ties on $Hu = 0$). In particular, for $v \in \mathcal{K}(D)$ we have the identity

$$[Hv]_+ = D Hv, \tag{4}$$

where $[\cdot]_+$ is taken entrywise.

## A.3 FROM THE CONVEX PROGRAM TO A TWO-LAYER RELU

Recall the sampled-pattern convex model (Eq. (3)):

$$\min_{\{(v_i, w_i)\}_{i=1}^P} \ell\left(\sum_{i=1}^P D_i H (v_i - w_i), y\right) + \beta \sum_{i=1}^P \big(\|v_i\| + \|w_i\|\big) \quad \text{s.t.} \quad v_i, w_i \in \mathcal{K}(D_i). \tag{5}$$

Here $D_i \in \mathcal{D}_H$ are (sampled) activation patterns. In the multi-class case, take $v_i, w_i \in \mathbb{R}^{d \times K}$ with columns $(v_{i,1}, \ldots, v_{i,K})$ etc., and interpret $\|\cdot\|$ as either the block $\ell_{2,1}$ norm, $\|M\|_{2,1} = \sum_{k=1}^K \|M_{:,k}\|_2$, or the Frobenius norm.

## A.4 VARIATION NORM AND A COMPUTABLE CERTIFICATE

We use the standard two-layer ReLU variation norm:

$$\|f\|_{\mathrm{var}} := \inf\Big\{ \sum_{j=1}^m \|a_j\|_2 \|u_j\|_2 : \ f(h) = \sum_{j=1}^m a_j [u_j^\top h]_+, \ m \in \mathbb{N} \Big\}.$$

The following result turns any feasible solution of equation 5 into an *explicit upper bound* on $\|f\|_{\mathrm{var}}$, hence a Lipschitz certificate for the logits.

**Proof of Theorem 2**

*Proof.* Using the representation in Proposition 2, build a two-layer network whose hidden units are the *columns* $\{v_{i,k}\}_{i,k}$ and $\{w_{i,k}\}_{i,k}$ with output weights $\{+e_k\}$ and $\{-e_k\}$ respectively. For each unit $(u, a)$ in this network, the atom cost is $\|a\|_2\|u\|_2 = \|u\|_2$ because $\|e_k\|_2 = 1$. Summing over units gives $\sum_{i,k} (\|v_{i,k}\|_2 + \|w_{i,k}\|_2) = \sum_i (\|v_i\|_{2,1} + \|w_i\|_{2,1})$, which upper bounds $\|f\|_{\mathrm{var}}$ by definition. For Frobenius penalties, $\sum_k \|M_{:,k}\|_2 \leq \sqrt{K}\|M\|_F$ yields the stated factor $\sqrt{K}$. $\qquad\square$

By Lemma 1, $\|f(h) - f(h')\|_\infty \leq \|f\|_{\mathrm{var}}\|h - h'\|_2$; combining with Theorem 1 yields the computable bounds

$$\mathrm{mar}(h + \delta, y) \geq \mathrm{mar}(h, y) - 2\widehat{\mathcal{B}}_{\mathrm{cvx}}^{(2,1)}\|\delta\|_2, \quad \mathrm{mar}(h + \delta, y) \geq \mathrm{mar}(h, y) - 2\sqrt{K}\,\widehat{\mathcal{B}}_{\mathrm{cvx}}^{\mathrm{F}}\|\delta\|_2,$$

depending on which penalty is used in Eq. (3). If the encoder $E$ is $L_E$-Lipschitz, replace $\|\delta\|_2$ by $L_E\|x - x'\|_2$ to get the end-to-end certificate.

## A.5 AM–GM LINK TO THE NONCONVEX $\ell_2^2$ PENALTY

Consider the two-layer model $f(h) = \sum_{j=1}^m a_j[u_j^\top h]_+$ trained via the nonconvex penalty of Eq. (2): $(\beta/2)\sum_{j=1}^m (\|a_j\|_2^2 + \|u_j\|_2^2)$. By AM–GM, $2\|a_j\|_2\|u_j\|_2 \leq \|a_j\|_2^2 + \|u_j\|_2^2$, hence

$$\|f\|_{\mathrm{var}} \leq \frac{1}{2}\sum_{j=1}^m (\|a_j\|_2^2 + \|u_j\|_2^2) = \frac{1}{\beta}\frac{\beta}{2}\sum_{j=1}^m (\|a_j\|_2^2 + \|u_j\|_2^2).$$

Therefore any solution of Eq. (2) yields the certificate

$$\|f\|_{\mathrm{var}} \leq \frac{1}{\beta}\mathcal{R}_{\ell_2^2} \quad\Rightarrow\quad \mathrm{mar}(h + \delta, y) \geq \mathrm{mar}(h, y) - \frac{2}{\beta}\mathcal{R}_{\ell_2^2}\|\delta\|_2.$$

Larger $\beta$ tightens the bound linearly.

## A.6 DETAILS FOR THE LOGIT LIPSCHITZ BOUND

Let $f(h) = \sum_j a_j[u_j^\top h]_+$. Since $t \mapsto [t]_+$ is 1-Lipschitz,

$$|f_k(h) - f_k(h')| = \left|\sum_j a_{j,k}([u_j^\top h]_+ - [u_j^\top h']_+)\right| \leq \sum_j |a_{j,k}|\,|u_j^\top(h - h')| \leq \sum_j \|a_j\|_2\|u_j\|_2\,\|h - h'\|_2.$$

Taking $\max_k$ and the infimum over all representations yields $\|f(h) - f(h')\|_\infty \leq \|f\|_{\mathrm{var}}\|h - h'\|_2$, which is the Lemma used in Theorem 1.

# B RELATED WORK CONTINUED

**Convex Reformulations of Neural Networks.** Deep learning optimization is traditionally nonconvex and largely heuristics-driven, yet it is possible to establish exact convex reformulations for two-layer neural networks using gated ReLU patterns (Pilanci & Ergen, 2020; Bach, 2017; Bengio et al., 2005). These reformulations offer polynomial-time convergence to global optima and have been extended to various architectures, including CNNs and Transformers (Ergen & Pilanci, 2021; Sahiner et al., 2021). However, prior work has largely focused on theoretical properties or small-scale image benchmarks (e.g., CIFAR-10). More recently Feng et al. (2024) have demonstrated the successful application of convex networks on text-input LLM tasks. Our work scales convex training to handle higher-dimension and complex speech representations, demonstrating that convex guarantees can translate to tangible performance gains in real-world large-scale spoken dialogue systems.

**Training Data Scarcity.** Contemporary studies (Reitmaier et al., 2022) have focused on more clearly defining the challenges ASR models face with low-resource languages, such as Xhosa or Marathi. Limited training data is a dominant issue, and authors (Babirye et al., 2022) have worked on building partnerships to preserve and document valuable linguistic data by remotely engaging local participants to record themselves, identifying more recording opportunities, and categorizing challenges of ASR in deeply multicultural communities. This has uncovered valuable implications

for collaborations across ASR and Human Computer Interface (HCI) that advance important discussions, while collecting more diverse speech datasets. Although promising, this approach also brings up new questions on the ethics of analyzing community voice recordings through platforms such as WhatsApp (Barbosa & Milan, 2019), and is slow to provide clearly annotated data from numerous low-resource languages.

**GPU Accelerated JAX Methods.** The importance of GPU acceleration has fueled much of the success in contemporary optimization and deep learning. The strong parallelization and speed potential of this framework has been demonstrated in the optimization works of Gu et al. (2025), Li et al. (2025), and Xu et al. (2025). This motivates us to solve the convex reformulation with a batched, multi-GPU ADMM solver (Boyd et al., 2011), yielding rapid convergence and efficient use of modern accelerators. As a result, our open-source implementation is easy to reproduce and plug into existing ASR pipelines or other encoder–decoder architectures. This design also aligns CLD with recent JAX-based convex and LLM systems such as CRONOS (Feng et al., 2024), and opens a clear path for future work on scaling convex language detection to larger encoders, cloud TPUs, and broader multilingual speech and multimodal agent application settings.

## C EXTENDED EMPIRICAL RESULTS

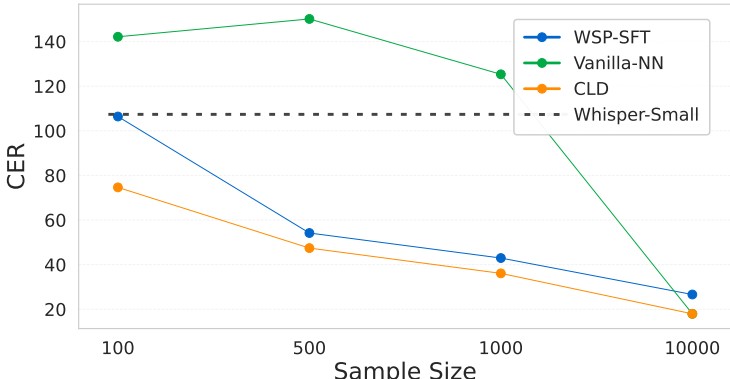

Figure 3: Character Error Rates (CER) of Whisper-small default, Whisper-small fine-tuned, vanilla-NN, and CLD on binary classification between English and Chinese across training sample sizes of 100, 500, 1000, and 10 000.

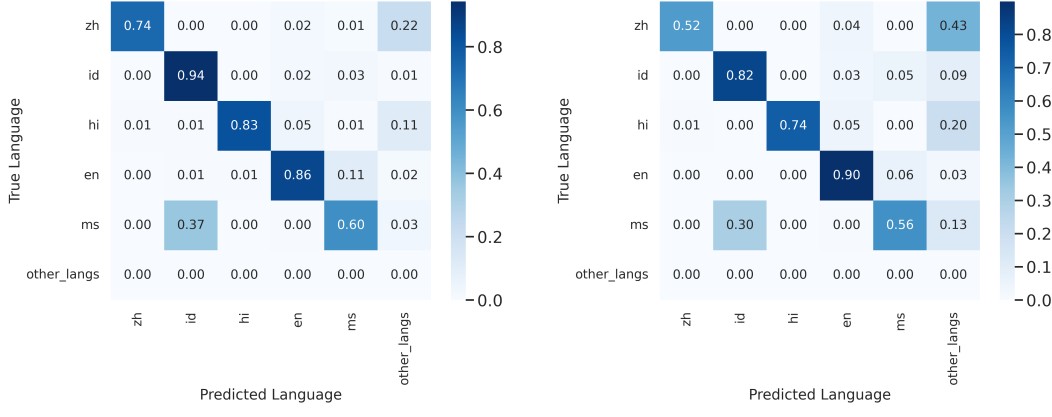

Figure 4: Language Classification Performance between WSP-SFT (left) and WSP (right) on Whisper-small across English (en), Chinese (zh), Indonesian (id), Malaysian (ms), Hindi (hi), and other predicted languages.

## C.1 Classification Accuracy per Sample Size Ablation

| Language | Accent / Dialect | Size | Correctly Predicted Samples | | | | Accuracy | | | |
|---|---|---|---|---|---|---|---|---|---|---|
| | | | WSP | WSP-SFT | NN | CLD | WSP | WSP-SFT | NN | CLD |
| English | Hindi-accented | 190 | 176 | 93 | 7 | 185 | 0.9263 | 0.9263 | 0.0368 | **0.9737** |
| English | Malaysian-accented | 215 | 136 | 137 | 7 | 213 | 0.6326 | 0.6372 | 0.0326 | **0.9907** |
| English | Singaporean-accented | 205 | 166 | 166 | 3 | 203 | 0.8098 | 0.8098 | 0.0146 | **0.9902** |
| English | Pakistani-accented | 189 | 182 | 182 | 10 | 186 | 0.9630 | **0.9630** | 0.0529 | 0.9841 |
| English | American | 204 | 195 | 195 | 12 | 203 | 0.9559 | 0.9559 | 0.0588 | **0.9951** |
| Chinese | Min Dong / Fuzhou | 71 | 7 | 7 | 67 | 71 | 0.0986 | 0.0986 | 0.9437 | **1.0000** |
| Chinese | Pu-Xian | 216 | 32 | 33 | 204 | 213 | 0.1481 | 0.1528 | 0.9444 | **0.9861** |
| Chinese | Hong Kong Cantonese | 184 | 121 | 121 | 175 | 175 | 0.6576 | 0.6576 | **0.9511** | 0.9511 |
| Chinese | Taiwanese Mandarin | 181 | 176 | 175 | 180 | 181 | 0.9724 | 0.9669 | **0.9945** | 1.0000 |
| Chinese | Mainland Mandarin | 205 | 187 | 187 | 201 | 199 | 0.9122 | 0.9122 | **0.9805** | 0.9707 |
| **Total** | | 1860 | 1378 | 1286 | 846 | 1829 | 0.7077 | 0.7102 | 0.3220 | **0.9742** |

Table 2: Multi-dialect classification accuracy between English and Chinese across 10 accents for (Whisper-baseline) WSP, WSP-SFT, vanilla-NN, and CLD at 100 samples.

| Language | Accent / Dialect | Size | Correctly Predicted Samples | | | | Accuracy | | | |
|---|---|---|---|---|---|---|---|---|---|---|
| | | | WSP | WSP-SFT | NN | CLD | WSP | WSP-SFT | NN | CLD |
| English | Hindi-accented | 190 | 18 | 175 | 190 | 185 | 0.9000 | 0.9211 | **1.0000** | 0.9737 |
| English | Malaysian-accented | 215 | 12 | 115 | 215 | 214 | 0.5714 | 0.5349 | **1.0000** | 0.9953 |
| English | Singaporean-accented | 205 | 16 | 154 | 205 | 203 | 0.8000 | 0.7512 | **1.0000** | 0.9902 |
| English | Pakistani-accented | 189 | 17 | 180 | 189 | 187 | 0.9444 | 0.9524 | **1.0000** | 0.9894 |
| English | American | 204 | 17 | 195 | 204 | 203 | 0.9444 | 0.9559 | **1.0000** | 0.9951 |
| Chinese | Min Dong / Fuzhou | 71 | 6 | 30 | 11 | 69 | 0.2727 | 0.4225 | 0.1549 | **0.9718** |
| Chinese | Pu-Xian | 216 | 3 | 82 | 47 | 216 | 0.2143 | 0.3796 | 0.2176 | **1.0000** |
| Chinese | Hong Kong Cantonese | 184 | 18 | 143 | 53 | 182 | 0.6667 | 0.7772 | 0.2880 | **0.9891** |
| Chinese | Taiwanese Mandarin | 181 | 15 | 181 | 33 | 181 | 0.8824 | **1.0000** | 0.1823 | 1.0000 |
| Chinese | Mainland Mandarin | 205 | 20 | 192 | 45 | 204 | 0.8696 | 0.9366 | 0.2195 | **0.9951** |
| **Total** | | 1860 | 122 | 1447 | 1192 | **1844** | 0.7066 | 0.7632 | 0.7062 | **0.9900** |

Table 3: Multi-dialect classification accuracy between English and Chinese across 10 accents for (Whisper-baseline) WSP, WSP-SFT, vanilla-NN, and CLD at 1000 samples.

| Language | Accent / Dialect | Size | Correctly Predicted Samples | | | | Accuracy | | | |
|---|---|---|---|---|---|---|---|---|---|---|
| | | | WSP | WSP-SFT | NN | CLD | WSP | WSP-SFT | NN | CLD |
| English | Hindi-accented | 190 | 176 | 180 | 187 | 180 | 0.9263 | 0.9474 | **0.9842** | 0.9474 |
| English | Malaysian-accented | 215 | 136 | 135 | 215 | 194 | 0.6326 | 0.6279 | **1.0000** | 0.9023 |
| English | Singaporean-accented | 205 | 166 | 166 | 204 | 195 | 0.8098 | 0.8098 | **0.9951** | 0.9512 |
| English | Pakistani-accented | 189 | 182 | 185 | 188 | 183 | 0.9630 | 0.9788 | **0.9947** | 0.9683 |
| English | American | 204 | 195 | 197 | 204 | 194 | 0.9559 | 0.9657 | **1.0000** | 0.9509 |
| Chinese | Min Dong / Fuzhou | 71 | 7 | 50 | 70 | 71 | 0.0986 | 0.7042 | 0.9859 | **1.0000** |
| Chinese | Pu-Xian | 216 | 32 | 198 | 215 | 216 | 0.1481 | 0.9167 | 0.9954 | **1.0000** |
| Chinese | Hong Kong Cantonese | 184 | 121 | 170 | 174 | 184 | 0.6576 | 0.9239 | 0.9457 | **1.0000** |
| Chinese | Taiwanese Mandarin | 181 | 176 | 181 | 181 | 181 | 0.9724 | **1.0000** | **1.0000** | **1.0000** |
| Chinese | Mainland Mandarin | 205 | 187 | 195 | 197 | 205 | 0.9122 | 0.9512 | 0.9610 | **1.0000** |
| **Total** | | 1860 | 1378 | 1657 | 1815 | 1803 | 0.7077 | 0.8026 | 0.9162 | **0.9721** |

Table 4: Multi-dialect classification accuracy between English and Chinese across 10 accents for (Whisper-baseline) WSP, WSP-SFT, vanilla-NN, and CLD at 10 000 samples.

# D LANGUAGE CLASSIFICATION ACCURACY OF BASELINE WHISPER ASR

Table 6: Whisper-small's default language detection accuracy by language and accent.

| Language | Accent | Samples | Correct | Accuracy (%) |
|---|---|---|---|---|
| **ID** | | | | |
| | Betawi | 505 | 451 | 89.3 |
| | Javanese | 182 | 163 | 89.6 |
| | Jawa Tengah | 228 | 206 | 90.4 |
| | Surakarta | 221 | 200 | 90.5 |
| | Bindeng | 221 | 200 | 90.5 |
| | Tionghoa | 221 | 200 | 90.5 |
| | Medhok | 224 | 203 | 90.6 |
| **EN** | | | | |
| | Malaysian English | 1000 | 614 | 61.4 |
| | Filipino | 1000 | 745 | 74.5 |
| | Singaporean English | 1000 | 752 | 75.2 |
| | Zimbabwe | 1000 | 831 | 83.1 |
| | Southern African (South Africa, Namibia) | 1000 | 831 | 83.1 |
| | Welsh English | 1000 | 918 | 91.8 |
| | Scandinavian | 1000 | 952 | 95.2 |
| | Pakistan | 1000 | 967 | 96.7 |
| | India and South Asia (India, Sri Lanka) | 1000 | 967 | 96.7 |
| | Scottish English | 1000 | 968 | 96.8 |
| | Lancashire English | 1000 | 970 | 97.0 |
| | Liverpool English | 1000 | 970 | 97.0 |
| | England English | 1000 | 982 | 98.2 |
| | Australian English | 1000 | 984 | 98.4 |
| | United States English | 1000 | 985 | 98.5 |
| | New Zealand English | 1000 | 987 | 98.7 |
| | Hong Kong English | 1000 | 988 | 98.8 |
| | German English | 1000 | 996 | 99.6 |
| | Non native speaker | 1000 | 996 | 99.6 |
| | Irish English | 1000 | 996 | 99.6 |
| | Canadian English | 1000 | 999 | 99.9 |
| | Northern Irish | 1000 | 999 | 99.9 |
| | Low | 1000 | 999 | 99.9 |
| | Demure | 1000 | 999 | 99.9 |
| | Midwestern | 1000 | 1000 | 100.0 |
| | Transatlantic English | 1000 | 1000 | 100.0 |
| **ZH** | | | | |
| | cdo | 1000 | 103 | 10.3 |
| | cpx | 1000 | 111 | 11.1 |
| | nan-tw | 1000 | 545 | 54.5 |
| | hk | 1000 | 905 | 90.5 |
| | zh | 1000 | 910 | 91.0 |
| | tw | 1000 | 987 | 98.7 |
| **HI** | | | | |
| | Kashmiri | 320 | 185 | 57.8 |
| | Bodo | 380 | 270 | 71.1 |
| | Malayalam | 365 | 271 | 74.2 |
| | Punjabi | 236 | 178 | 75.4 |
| | Urdu | 131 | 103 | 78.6 |
| | Telugu | 474 | 377 | 79.5 |
| | Tamil | 182 | 145 | 79.7 |
| | Hindi | 575 | 468 | 81.4 |

| Language | Accent | Samples | Correct | Accuracy (%) |
|---|---|---|---|---|
| | Sindhi | 141 | 116 | 82.3 |
| | Odia | 374 | 311 | 83.2 |
| | Kannada | 313 | 268 | 85.6 |
| | Gujarati | 299 | 257 | 86.0 |
| | Assamese | 262 | 226 | 86.3 |
| | Konkani | 422 | 364 | 86.3 |
| | Nepali | 359 | 311 | 86.6 |
| | Bengali | 418 | 366 | 87.6 |
| | Dogri | 219 | 194 | 88.6 |
| | Maithili | 287 | 255 | 88.9 |
| | Marathi | 395 | 366 | 92.7 |
| **MS** | | | | |
| | msi | 1000 | 386 | 38.6 |
| | ms | 1000 | 934 | 93.4 |

## E  HUMAN EVALUATION

| Method | Total Test Prompts | Wrong Language Transcribed | Word Errors in Transcription |
|---|---|---|---|
| *Default* | | | |
| EN | 595 | 59 | – |
| ZH | 300 | 148 | – |
| *vanilla-NN* | | | |
| EN | 450 | 22 | 81 |
| ZH | 450 | 5 | 14 |
| *CLD (ours)* | | | |
| EN | 450 | 12 | 26 |
| ZH | 450 | 2 | 14 |

Table 7: Human feedback comparison on Whisper-small using the Default, vanilla-NN, and CLD language classifiers. The tests were conducted on five humans in Singapore for English (EN) and ten humans in South-East China for Mandarin (ZH).

## F  ALTERNATING DIRECTION METHOD OF MULTIPLIERS

The Alternating Direction Method of Multipliers (ADMM) is an optimization algorithm that has gained significant prominence in solving large-scale convex optimization problems, particularly those involving decomposable objective functions and constraints.

**ADMM Background.** ADMM is an optimization framework designed to efficiently solve convex minimization problems that involve complex or non-smooth regularization terms. It works by breaking down a large global problem into a sequence of smaller, easier-to-solve local subproblems, which are then solved iteratively and linked together via Lagrange multipliers (the "multipliers" component) and a quadratic penalty term (the "augmented Lagrangian" component). This structure is ideal when the objective function is composed of several parts that apply to different subsets of variables, making the problem separable.

**Novelty and Significance.** In the context of Convex Language Detection (CLD), ADMM is critical and represents a significant novelty since it provides:

- **Tractability for Scale:** CLD is derived from a convex reformulation of a two-layer ReLU network. While theoretically sound, solving the resulting optimization problem (Eq. 2) can be extremely challenging for high-dimensional data, such as the extracted ASR hidden

features. ADMM makes this problem tractable by allowing the optimization to be performed in a batched, multi-GPU parallel fashion (especially when implemented in JAX). ADMM is perfectly suited for solving such problems by isolating the non-smooth term in one subproblem (the proximal update).

- **Guaranteed Global Optimality:** By solving the convex program using a principled method like ADMM, the training process is guaranteed to converge to the unique global optimum. This eliminates the dependency on brittle hyperparameters (like learning rates) and avoids the local minima issues that plague traditional neural network training (e.g., the standard MLP baseline).

- **Novelty in ASR/Speech:** While ADMM is well-known in optimization (Boyd et al., 2011), we assert that its practical application as a core training method for complex deep learning reformulations on large-scale spoken dialogue systems remains highly novel. This innovation allows CLD to achieve massive compute efficiency (requiring 13x fewer TFLOPs than the standard vanilla-NN) and drastically reduced training time.

