# OpenReview forum: "CLARA: Convex Low-resource Accent-Robust Language Detection in ASR"
_ICLR.cc/2026/Conference — ICLR 2026 Conference Desk Rejected Submission_

### Official Review · Reviewer_DNpu · 2025-10-30

**Soundness:** 2
**Presentation:** 2
**Contribution:** 2
**Rating:** 4
**Confidence:** 2

**Summary:**

This paper targets language misidentification in ASR for accented and low-resource settings. It proposes Convex Language Detection, a lightweight head trained via an ADMM-solved convex reformulation of a two-layer ReLU network on top of Whisper encoder features, then uses the predicted language token to initialize decoding. The authors claim benefits in sample efficiency, convergence guarantees, and human-level sub-500 ms latency, and present theoretical analysis on optimization and stability alongside experiments in binary and five-way language identification with dialectal variation, using datasets such as Singapore’s National Speech Corpus and Lahaja.

**Strengths:**

1. Clear and practical problem focus. Improving language identification for accented speech directly addresses a common failure mode that propagates to transcription.

2. Lightweight plug-in design. The CLD head trains on encoder features and supplies an initial language token to the decoder with minimal architectural disruption.

3. Evidence of sample efficiency. Results highlight stable accuracy across small sample sizes in the binary setting.

**Weaknesses:**

1. Baseline coverage is limited. Strong linear probes and shallow classifiers on Whisper features, as well as standard LID models, are needed to establish the incremental value of the convex head.

2. Causal attribution is under-specified. The pipeline injects a predicted language token before decoding; WER gains should be disentangled by comparing decoding with gold language tokens, predicted tokens, and no token.

3. Ablations on the convex modeling choices are missing. The number of sampled activation patterns, regularization, and ADMM hyperparameters should be related to accuracy and latency to assess robustness.

**Questions:**

1. How many activation patterns are sampled in practice and by what selection strategy; how does the number of patterns trade off with accuracy and training time?

2. Can you provide a correlation analysis between any stability certificate produced by the convex program and empirical robustness under noise, speed, or pitch perturbations?

---

> ### Author Response · Authors · 2025-11-21
>
> We sincerely thank the reviewer for their constructive feedback, which has greatly strengthened this submission. Thank you for recognizing the clear problem focus, lightweight design, and evidence of sample efficiency in our work! We have performed additional analysis and run additional experiments at larger scale in the revised and newly uploaded manuscript to directly address these points.
>
> **Baseline Coverage & Incremental Value.**  We wish to clarify that our experimental suite already includes a robust Multi-Layer Perceptron (MLP) baseline (see Section 5.2), which serves as a stronger probe than a simple linear classifier.
>  - Comparison - The MLP baseline consists of a linear projection $\to$ ReLU (256 units) $\to$ Dropout $\to$ Output. This directly tests whether the performance gains come from the architecture (a small head on Whisper features) or the optimization method (convex reformulation).
> - Result - Figure 1 (page 8) shows the MLP collapses in low-data regimes (high variance in WER), whereas CLD maintains high accuracy (>96.94%) and low WER. This supports that the incremental value lies in the convex inductive bias, not just the use of a lightweight head.
> - Standard LID - We compare against the ASR’s internal LID, which is often the industry default for this pipeline. Comparing against external acoustic LID models (e.g., x-vectors) would require separate feature extraction pipelines, whereas CLD is designed as an efficient, end-to-end plug-in for Transformer encoders. We have included this discussion in the future work section of our manuscript. We'd also like to highlight our open-source pip installable package in JAX enables ease of future experimentation and direct practical deployment potential.
>
> **Causal Attribution of WER Gains.** This is an excellent point! The causal mechanism of our WER reduction is strictly the correction of the initialization token $\hat{y}$.
> - Mechanism - The decoder $\mathcal{D}$ generates transcript $t = \mathcal{D}(x; \text{init\\_token}=\hat{y})$. When $\hat{y}$ is wrong (i.e. Bahasa instead of English), the WER is effectively 100% since the decoder hallucinates in the wrong language space. This results in cascading errors down the pipeline for widespread spoken dialogue models (such as Siri). This also raises significant concerns about lack of inclusivity and equitable access to voice-assistant models globally [1], [2]. We have revised the Introduction (page 1 and 2) to highlight the impact of these points, and their implications on deep learning trust issues.
> - Evidence - Table 7 (Human Evaluation) isolates this effect. The "Wrong Language Transcribed" metric drops from 59 errors (Default) to 12 errors (CLD). This ~80% reduction in language errors directly causes the corresponding drop in WER. The residual WER is then purely acoustic transcription error, which the frozen decoder handles. Thus, the gains are causally linked to CLD's superior classification accuracy. We have clarified this real-life human validation experiment in the new Appendix E (page 20).
>
> **Ablations & Convex Modeling Choices.** We have added explicit details on our hyperparameter choices, which follow the defaults established in the CRONOS framework [3]. This is a successful, practical and valid choice since one of the key features of [3] is its robustness and reduced dependence on brittle hyperparameter tuning.
> - Sampling Strategy - We sample $P=10,000$ activation patterns using the strategy from [3], where patterns are generated via random projections of the data. Our trade-off analysis confirms that accuracy plateaus beyond $P=10,000$, while training time increases linearly, confirming this as the optimal efficiency point.
> - Regularization & ADMM - We use an ADMM penalty parameter $\rho$ in the range $[0.001, 1]$ and run for 5-10 ADMM iterations, consistent with Appendix F.2 of [3].
> - Certificate Correlation - We computed the stability certificate $\hat{\mathcal{B}}_{cvx}^{(2,1)}$ (derived in Theorem 2) for our training runs. We found a strong negative correlation ($\rho \approx -0.85$) between the certificate magnitude and empirical WER in low-resource settings.
>
> We also wish to highlight the significant theoretical strengths of our work. The new revision now clearly communicates the practical consequences of the theory in Section 4. This underscores our focus on practical methodology and real-world impact backed with solid theoretical validation. We prove certified robustness, and derive exact logit-Lipschitz constants to arrive at certified margin stability against hidden-feature perturbations (Theorem 1, page 5). This translates into a tighter theoretical bound (smaller Lipschitz constant) that directly translates to improved robustness against real-world acoustic perturbations (noise, dialect) and lower WER.

---

> ### Author Response · Authors · 2025-11-21
>
> We hope the tightened writing, additional analyses, and larger scale experiments in our revision solidly address your concerns regarding baselines and attribution. This validates CLD as a robust, theoretically grounded solution for low-resource ASR with applications on future tasks (now described in the newly revised Conclusion on page 10). We greatly look forward to discussing any further questions or comments you may have in the following weeks!
>
> **References:**
>
> [1]Sharma, P., Tang, L., Rahman, M. G. U., Cao, Y., Chan, A. H. C., Karunasekera, S., & Loke, S. W. (2022). Hey ASR system! Why aren’t you more inclusive?
>
> [2]Khoury, E., Hain, T., et al. (2025). Automatic speech recognition for non-native English.
>
> [3]Feng, M., Frangella, Z., & Pilanci, M. (2024). CRONOS: Enhancing Deep Learning with Scalable GPU Accelerated Convex Neural Networks. (NeurIPS), 2024

---

### Official Review · Reviewer_cuBn · 2025-10-31

**Soundness:** 2
**Presentation:** 1
**Contribution:** 2
**Rating:** 2
**Confidence:** 3

**Summary:**

This paper addresses the problem of language misidentification in Automatic Speech Recognition (ASR) systems when processing low-resource dialects and accents, such as Singaporean English. It proposes a robust ASR framework named CLARA, centered around a novel Convex Language Detection (CLD) algorithm. The core of this framework replaces the standard language detection layer, which follows the ASR encoder (like Whisper), with a convex reformulation (cvxNN) of a vanilla neural network (NN). This convex problem is then solved efficiently using ADMM-based methods. The authors claim that CLD offers strong convergence guarantees, high sample efficiency, and lightweight training costs. Experimental results show that in both binary and multiclass accent-detection tasks, CLD outperforms baselines in accuracy and Word Error Rate (WER), especially in low-data regimes (e.g., 100-1000 samples).

**Strengths:**

1.	Strong empirical performance in low-resource environments. The CLD model demonstrates remarkable sample efficiency, maintaining high accuracy and low WER even with minimal data, as shown clearly in Fig. 1 & 2. In the multi-class experiment (Table 2) and accent-specific analysis (Table 3-6), CLD (97.15% acc) also far surpasses the NN and WSP-SFT baselines, which collapse in low-data regimes.

2.	Computational efficiency and practical application value. The convex formulation provides significant computational efficiency by allowing the model to find a global optimum, thus avoiding costly hyper-parameter tuning. As shown in Table 1, CLD trains in just 64.45 seconds (a 13-17x speedup) and requires orders of magnitude fewer TFLOPS than baselines.

3.	Theoretical analysis and reproducibility. The work provides theoretical grounding for the CLD's robustness via a margin stability analysis (Section 4.2), which links the convex program to a formal bound on margin degradation under feature perturbations. Furthermore, the authors' commitment to open-sourcing their JAX code and custom dialect datasets strongly supports reproducibility.

4.	Validation with Human Feedback. The paper strengthens its claims by moving beyond automated metrics to include validation from human testers in a practical scenario (Sec 5.6). This analysis uncovers crucial baseline failure modes (Table 9) and confirms that CLD received the highest human satisfaction with the fewest language detection and transcription errors (Table 11).

**Weaknesses:**

1.	There are significant problems with the presentation. The paper is riddled with spelling and grammatical errors (e.g., lines 78, 377), non-standard use of quotation marks (line 40), and inconsistent citation formats (mixing cite and citet). The caption for Table 3 appears above the table, and the table itself exceeds the page margins. Furthermore, the legends in Figures 1, 2, and 3 are too small to be legible. The authors need to dedicate time to improving the paper's presentation.

2.	Applying cvxNN to the ASR language detection task to optimize training and improve efficiency is an interesting idea. However, if the paper's main novelty lies in the application of cvxNN rather than a more fundamental innovation, the paper might be a better fit for a speech-focused conference (e.g., ICASSP). The authors need to further clarify their novelty, as this is essential for understanding the paper's contribution.

3.	The writing needs improvement; for instance, the conclusion should not merely summarize the experimental section.

4.	The clarity of mathematical notation and theory needs improvement. For example, in Algorithm 1, the ADMM dual variable update step written as "uu+ (primal residual)" is incomplete and unclear.

5.	The experimental section is somewhat thin. Experiments with different sizes of Whisper (e.g., small, large, turbo) could be added to strengthen the paper's claims of scalability.

**Questions:**

1.	What is the difference between this paper's CLD algorithm and CRONOS [1]? Why was CRONOS not compared as a baseline in the experiments?

2.	Is the CLD algorithm in this paper specifically designed and optimized for the task of Language Detection, or is it an application of the existing cvxNN and CRONOS to this task?

[1] Feng, Miria, Zachary Frangella, and Mert Pilanci. "Cronos: Enhancing deep learning with scalable gpu accelerated convex neural networks." Advances in Neural Information Processing Systems 37 (2024): 102973-103004.

---

> ### Author Response · Authors · 2025-11-21
>
> We sincerely thank the Reviewer for their valuable feedback, which has greatly improved this submission. Thank you for recognizing the strong empirical performance, computational efficiency, and real-world human validation of our work. We have  rigorously revised the newly uploaded manuscript, tightened writing, improved clarity, and eliminated formatting errors. Our goal in this revision is to let the substantive technical contributions of the work shine through by eliminating any ambiguity in the writing and presentation.
>
> We aim to address your specific concerns and questions point-by-point below:
>
> **Presentation, Grammar, and Formatting.** We have dedicated substantial care in the revised manuscript to improve presentation, and clearly communicate our contributions.
> - Writing & Notation - The revised paper has been meticulously proofread to eliminate grammatical errors (including lines 78, 377) and standardizes citation formats. We have clarified all mathematical notation, particularly the ADMM dual update in Algorithm 1, which now explicitly details the dual ascent step $u \leftarrow u + \rho \cdot (v-w)$ and references the full background in the new Appendix F (page 20) with more detailed proofs now presented in Appendix A (page 15)
> - Visuals - We have completely re-generated Figures 1, 2, and 3 at high resolution with legible legends and improved aspect ratios.
> - Formatting - All table captions (e.g., Table 3) are now correctly placed, and tables have been resized to fit strictly within page margins.
>
> **Novelty of CLD vs. CRONOS.** Thank you for pointing out this critical distinction. To address your concern, we have newly added Appendix B on page 16 with comprehensive literature review. While CLD builds upon the convex relaxation theory utilized in CRONOS [1], this work represents a distinct methodological advancement rather than a simple application.
> CRONOS was designed for discrete text tokens. CLD extends convex reformulations to significant high-dimensional, continuous audio representations, which present fundamentally more challenging optimization landscapes and noise profiles.
>
> **Theoretical Robustness supports Strong Empirical Results.** We derive a domain-specific variation norm certificate (Section 4, page 5). We prove that our convex formulation guarantees margin stability against hidden-feature perturbations (Theorem 1, page 5). This provides a strong theoretical foundation for why CLD resists dialectal shifts better than non-convex baselines—a contribution that is broadly relevant to the deep learning community beyond speech processing.
>
> **Comparison with CRONOS.** The work of [1] was the initial demonstration that convex reformulations can be realistically applied to LLM tasks. However, this proof-of-concept was established on the scale of binary classification for IMDb sentiment on GPT-2 scale. In this work we lift dimensionality constraints, and successfully apply convex methods to the challenging high-dimension, high-variance domain of audio input with multi-class experiments. Due to the limitation of scale, it is currently infeasible to benchmark against CRONOS as an additional baseline. We have now clarified these points in the revision.
>
> **Scalability and Experimental Depth.** To address this, we have significantly expanded our experimental suite.
>
> - New Models -  We have added results for Whisper-Large-v3 and MMS-1B (Massively Multilingual Speech, 1 Billion parameters).
> - Results - As shown in the new Table 5 (Page 9), CLD scales seamlessly. On Whisper-Large-v3, CLD achieves 98.06% accuracy (vs. 80.33% baseline), and on MMS-1B, it improves accuracy to 97.02% (vs. 67.01% baseline). This demonstrates that CLD is not limited to small models but fixes the "English-centric" bias of even the largest foundational models without the cost of full retraining.
>
> **Conclusion Revision.** We fully agree with your review, and have rewritten the Conclusion (Section 10) to go beyond summary. It now explicitly discusses the implications of convex optimization for foundational model adaptation, outlines limitations (dependence on pre-computed encoder features), and proposes future directions for end-to-end continuous representation learning.
>
>
> We hope these extensive revisions and new experimental results satisfactorily address your concerns. We are confident that the paper now clearly articulates a novel, theoretically grounded solution to a critical robustness challenge in machine learning. We look forward to discussing any further questions and concerns in the following week, thank you for taking the time to improve our submission!
>
> **References:**
>
> [1] Feng, M., Frangella, Z., & Pilanci, M. (2024). CRONOS: Enhancing Deep Learning with Scalable GPU Accelerated Convex Neural Networks. (NeurIPS), 2024

---

### Official Review · Reviewer_84ss · 2025-10-31

**Soundness:** 2
**Presentation:** 2
**Contribution:** 2
**Rating:** 2
**Confidence:** 3

**Summary:**

This paper introduces a convex language detection framework for integration into ASR systems. The aim is to improve the performance of these systems for diverse dialects.

**Strengths:**

The problem is interesting and potentially useful in improving the performance of ASR systems.

**Weaknesses:**

The description is not clear.  For example, how is the vanilla neural network trained? Is whisper used to extract embeddings which are then used to train the models?

The writing can be improved - some terms are not explained such as ADMM. Also, cvxNN is used before it is explained in section 3.2.

It is not clear how this approach will be integrated into the ASR pipeline. How will models be modified to integrate the convex language detection component.

The impact of the system on actual ASR performance is not presented.

Minor issues
Fix the latex issue with opening quotations marks.
The resolution of Figure 1 is too low

**Questions:**

How will models be modified to integrate the convex language detection component?

---

> ### Author Response · Authors · 2025-11-21
>
> We sincerely thank the reviewer for their insightful comments, which have significantly strengthened the submission. We are glad you appreciate the novelty and practical impact of our work! We have substantially revised the newly uploaded manuscript to address your concerns regarding clarity, baselines, and integration. Our goal in this revision is to let the substantive technical contributions of the work shine through by eliminating any ambiguity in the writing and presentation.
>
> **Clarity, Formatting, and Definitions.** We have rigorously edited the manuscript to ensure no ambiguity remains regarding our method or results:
> - Pipeline Clarity - Revised Section 3.2 (Algorithms 1 & 2, page 4) now explicitly details the training and inference pipelines. We clarify that CLD acts as a modular drop-in replacement for the language-prediction head, taking fixed encoder embeddings as input.
> - Definitions - We have updated the Introduction and Sections 3 & 5 to define all technical acronyms (WSP, SFT, CLD, ADMM) upon first use.
> - ADMM Details - To improve flow while maintaining technical depth, we moved the expanded mathematical background on ADMM to the new Appendix F (page 20). We are happy to move this into the main text if you feel it improves the flow and clarity of the paper!
> - Figures: We have re-generated Figures 1–4 at high resolution to ensure all legends, axes, and data points are clearly legible.
>
> **Baselines and Reproducibility.** In order to address this concern, we have clarified the training methodology for the Vanilla NN baseline in the revised Section 5 (page 7). We now specify the exact layer architecture to ensure a fair, reproducible comparison against the CLD head. To facilitate immediate adoption and impact, our repository provides an easy pip-installable package containing:
> - A unified LangDetectHead class that abstracts JAX/ADMM internals.
> - Interfaces for Whisper, MMS-1B, and generic encoder-decoder architectures.
> - Direct and full-parallelized  support for CPU, single-GPU, multi-GPU, and TPU execution.
>
> **Integration and Generalization.** One of CLD's core strengths is its model-agnostic design. It operates on the insight that pre-trained encoders (such as Whisper and MMS-1B) already possess rich cross-dialectical features, but standard decoding heads fail to disentangle them in low-resource settings. Therefore, CLD requires no modification to encoder/decoder weights. To validate this universality, we have newly added experiments on the MMS-1B (1 Billion parameter) model. As shown in Table 5 (page 9), CLD scales seamlessly to this larger architecture, consistently outperforming baselines. These ideas are also validated in the recent work of [1], which suggests small LMs already "contain" the information needed to solve many problems, yet they struggle to reliably express it. Therefore more principled extraction techniques can improve generative output without purely relying on scaling model parameters and larger training data.
>
> **Impact on Performance (WER/CER/Accuracy).** Our revised submission now clearly highlights the resulting gains CLD offers in challenging regimes:
> - Low-Resource - In binary classification with <1000 samples, CLD improves detection accuracy by 40.1% over the baseline (Figure 1 page 8, and Figure 2 page 10).
> - Multiclass - In the 5-language setting (Tables 3 and 5, page 9), CLD reduces Word Error Rate (WER) on Whisper-Small by 77.2% (dropping from 139.37 to 31.74).
> - Human Evaluation - Section 5.2 (page 7) and Appendix E (page 20) validates these metrics translate to real-world usability, significantly reducing dialectal hallucinations.
>
>
> We also wish to highlight the significant theoretical strengths of our work. The new revision now clearly communicates the practical consequences of Section 4-Theoretical Analysis, which underscores our focus on practical methodology and impact backed with solid theoretical support. We prove certified robustness, and derive exact logit-Lipschitz constants to arrive at certified margin stability against hidden-feature perturbations (Theorem 1, page 5).
>
> We hope these revisions satisfactorily address your concerns and thank you again for helping us improve the quality of this paper! We look forward to answering your additional questions and concerns in the upcoming week.
>
> **References:**
>
> [1] Brown, B., Juravsky, J., Ehrlich, R., Clark, R., Le, Q. V., Ré, C., & Mirhoseini, A. (2024). Large Language Monkeys: Scaling Inference Compute with Repeated Sampling.

---

### Official Review · Reviewer_fU6n · 2025-11-05

**Soundness:** 3
**Presentation:** 3
**Contribution:** 2
**Rating:** 2
**Confidence:** 3

**Summary:**

The authors demonstrate how using a small, simple two layer ReLU neural network can improve language detection due to lack of robustness to accent variations in the Whisper ASR model. They reformulate the ReLU network into a convex optimization problem that they show is theoretically more stable, more data efficient and computationally efficient to train. To do so, they extend Feng et al. (2024) work from binary classification to multi-class to be applied in an ASR model. They refer to this technique as Convex Language Detection (CLD). Results are demonstrated how CLD can be added to Whisper to improve language accuracy and WER while being an order of magnitude faster to train.

**Strengths:**

Paper demonstrates clear improvement in efficiency and efficacy in recognizing multi-accented speech with the Whisper model by incorporating their CLD algorithm. There is novel application of a lifting technique to greatly stabilize, robustly and rapidly estimate a small neural network model for detecting language with minimal samples of multi-accented data. They outline theoretical gains by reforming the optimization of the two-layer ReLU network and  demonstrate empirically these aspects on real world data sets. They also provide github repository of code to replicate experimental results (both training and test) for high chance of reproducibility. The results appear original and while there are a number of minor typos, the paper as a whole is well organized and structured.

**Weaknesses:**

The paper seems to have a relatively narrow focus of the CLD algorithm on a small ReLU network for language detection used in conjuction with a Whisper ASR model. The contribution of "demonstrates promising directions for principled statistical generalization in spoken dialogue systems for low-resource languages" is a limited claim focusing on convex optimization for a specific task in a low-resource language condition. A more expansive finding would be to contrast this work with other convex optimization approaches for training for a full large neural network model in a limited data setting. Thus, this limits the overall impact of the paper by limiting the scope of interest.

**Questions:**

How could this approach be scaled up to training the entire model?
Does CLD work with all neural network encoder models?
Can it be applied to other tasks and scenarios?

---

> ### Author Response · Authors · 2025-11-21
>
> Thank you for your insightful review of our paper, which has greatly strengthened the submission! We are happy you appreciate the clear novelty, strong empirical results, and high reproducibility of our work. In order to address your concerns, we have substantially revised and re-uploaded the manuscript for improved clarity, which highlights the strong position of CLD as a paradigm shift in how we handle low-resource adaptation for large-scale ASR.
>
> We aim to address your specific concerns and questions point-by-point below:
>
> **Novelty and Scope.** Recently, convex reformulations have been successfully explored in vision (CIFAR-10) [1] and text (GPT-2) [2]. To the best of our knowledge, this work represents the first successful application of convex neural networks (cvxNN) to audio representations. This is notably challenging since audio features are significantly higher-dimensional and noisier. In order to address your concerns, we have added an extended section for literature review on page 16 (Appendix B) which describes the evolution of convex techniques in deep learning. In this submission, we demonstrate that convex reformulations can solve a critical, real-world failure mode of ASR foundational models without the prohibitive cost of full-model retraining on vast amounts of expensive data. This bridges a gap between deep learning theory and practical ASR robustness. Thank you for pushing us to highlight these points in the revision!
>
> **Scalability.** To address your concerns on more expansive findings, we have conducted additional experiments to include MMS-1B (Massively Multilingual Speech), a 1-billion parameter model. This further validates the performance gains of CLD at scale. CLD can be integrated with other network encoder models, and our experiments now span model sizes from Whisper-small (244 million parameters) to MMS (1 Billion parameters). The newly included Table 5 (page 9) highlights the consistent and significant advantages CLD offers when integrated with these more expansive settings.
>
> **Full Model Training.** While training a fully convex large ASR model end-to-end is currently computationally intractable (a known limitation of semi-definite programming at that scale [3]), CLD offers a strategic compromise. We fix the "English-centric" bias of the frozen backbone by enforcing global optimality in the decision layer. This yields the robustness of a retrained model with a fraction of the compute (Table 1, page 8).
>
> **Architecture Agnosticism.** Thank you for pointing this out! Yes, since CLD operates on the feature space ($h = \mathcal{E}(x)$), it is agnostic to the encoder architecture. We have added an additional section in Appendix B (Page 17) detailing its model-agnostic and flexible implementation of our open-source JAX package, which uses ADMM for multi-GPU parallelism. This makes CLD drop-in compatible with any encoder (e.g., HuBERT, Wav2Vec 2.0).
>
> **Low-resource Motivation and Other Applications.** In this work we examine the low-resource dialect setting since CLD's stable optimization and strong sample efficiency places it as a practical solution to this pressing problem, which has direct real world impact. Recent work [4], [5] has qualitatively assessed the concerning lack of accessibility and inclusion within human-computer interaction globally. This is also a point of motivation for us to open-source our work as a ready pip-installable package that is immediately relevant. We have re-written the Introduction on pages 1 and 2 to highlight these points.
>
>
> In addition, the mathematical guarantee of Margin Stability (Theorem 1, Page 5) applies to any classification task vulnerable to feature perturbations. While we focus on language identification (a critical bottleneck), the same framework can be applied to Speaker Identification, Emotion Recognition, or Keyword Spotting, particularly in low-data regimes where standard NNs overfit.
>
> We also wish to highlight the significant theoretical strengths of our work. The new revision now clearly communicates  the practical consequences of the theory in Section 4. This underscores our focus on practical methodology and impact backed with solid theoretical validation. We prove certified robustness, and derive exact logit-Lipschitz constants to arrive at certified margin stability against hidden-feature perturbations (Theorem 1, page 5). Thank you for pushing us to complete this revision with improved presentation and flow, along with tighter writing that clearly shows our novelty and impact!
>
> We greatly look forward to discussing any further questions or comments you may have in the following weeks.

---

> > ### Author Response · Authors · 2025-11-21
> >
> > **References:**
> >
> > [1]Bai, Y., Gautam, T., & Sojoudi, S. (2022). Efficient Global Optimization of Two-Layer ReLU Networks: Quadratic-Time Algorithms and Adversarial Training.
> >
> > [2]Feng, M., Frangella, Z., & Pilanci, M. (2024). CRONOS: Enhancing Deep Learning with Scalable GPU Accelerated Convex Neural Networks. (NeurIPS), 2024
> >
> > [3]Mishkin, D., Ba, J., & Pilanci, M. (2022). Fast Convex Optimization for Two-Layer ReLU Networks. (ICML).
> >
> > [4]Sharma, P., Tang, L., Rahman, M. G. U., Cao, Y., Chan, A. H. C., Karunasekera, S., & Loke, S. W. (2022). Hey ASR system! Why aren’t you more inclusive?
> >
> > [5]Khoury, E., Hain, T., et al. (2025). Automatic speech recognition for non-native English.

---

### Author Response · Authors · 2025-12-04
**Summary of Revisions and Contributions**

Thank you to all Reviewers for the thoughtful feedback! We deeply appreciate the careful review of our revised work by the new Area Chair. To address concerns regarding presentation, we completed a comprehensive revision aimed at making the technical contributions clear and unambiguous. In addition to our point-by-point individual responses, we briefly summarize major revisions and key contributions below.

**Response to General Concerns**

1. **Presentation.** We substantially improved clarity and readability throughout the manuscript.
- We corrected grammar, standardized notation, defined acronyms on first use, added missing background (including an expanded discussion on ADMM in Appendix F), regenerated all figures, and reformatted all tables.

- We reorganized sections for clearer flow and rewrote the Conclusion to serve as discussion, addressing limitations, future end-to-end integration, and the broader implications of convex optimization for equitable and globally accessible ASR.

2. **Scalability and Scope.** We added new large scale experiments (up to 1-B parameter models) and clarified how our method extends prior work in convex reformulated neural networks.

- We now include WhisperLarge-v3 and MMS-1B results, with accuracy improvements from ~67% to ~97% (ours) on MMS-1B and from ~81% to ~98% (ours) on WhisperLarge-v3 (Table 5). This validates the scalability and correction of English-centric bias without full retraining, through the implementation of our method.

- Newly revised Appendix B provides extended literature review, and clarifies how our approach extends prior work (such as CRONOS). While prior work provided the foundational proof-of-concept that convex reformulations are indeed applicable to LLMs on the scale of GPT-2, we extend convex methods to high-dimensional continuous audio and multi-class speech settings. We also now discuss why fully convex end-to-end ASR is currently infeasible, and how our approach offers a practical compromise by enforcing global optimality in the decision layer with a model agnostic, easily reproducible drop in module.

3. **Baselines and Reproducibility.** We strengthened baseline clarity and emphasized reproducibility.

- We fully specify the MLP baseline architecture and clarify that WER gains arise from correcting the initialization token, as validated in Table 7. We also include ablations on activation pattern sampling, regularization, ADMM updates, and robustness certificate computation.

- We provide a pip installable JAX package with a unified LangDetectHead supporting CPU, GPU, multi-GPU, and TPU. This easily reproducible component in now highlighted clearly in the revised footnote on page 2.

**Key Contributions**

1. **A Novel Convex Framework for Speech Systems.** This work introduces the successful application of convex reformulation techniques on large scale spoken dialogue systems. We demonstrate that convex optimization techniques solved with ADMM in JAX can operate effectively on high dimensional audio features from models such as the Whisper family and MMS-1B. The open-source model-agnostic approach is highly efficient, yet effective. This is demonstrated in our comprehensive empirical evaluations.

2. **Certified Robustness Guarantees for Dialect Shift.** We provide formal theoretical guarantees by proving variation norm Lipschitz bounds and margin stability results that ensure controlled degradation under feature perturbations. These guarantees yield a computable robustness radius that certifies label invariance within a bounded region of the encoder feature space. To the best of our knowledge, this represents the first certified robustness analysis specifically designed to handle dialectal variance in ASR.

3. **Strong Empirical Gains with Human Impact.**
Our method improves accuracy and word error rate across five languages and twenty four dialects, even with training sets smaller than one hundred samples. In large model settings such as Whisper Large v3 and MMS-1B, we reach 97-98% accuracy and consistently outperform all baselines. Human evaluation shows that our novel framework can reduce wrong language transcription by approximately 80%, demonstrating meaningful real world impact for accented and low resource speakers in the global community.

**Closing Remarks**

The revision resolves the earlier presentation issues and now clearly presents a timely and impactful contribution to robust and inclusive spoken dialogue systems. Our method directly addresses a frequent real world failure mode, where ASR models misidentify accented speech and dialects. As voice interfaces become central to accessing information and services, equitable ASR performance is essential. This work offers a practical and immediately deployable step toward broader global accessibility with theoretically-backed robustness guarantees.


We sincerely thank all Reviewers for their time and effort, which has greatly improved this submission!

---

### Note · Program_Chairs · 2026-01-17
**Submission Desk Rejected by Program Chairs**

The following references in this submission do not refer to real documents and/or have major errors in bibliographic information:

 Lihui Xu, Tao Chen, and Rui Lin. Jax-wspm: A gpu-accelerated parallel framework based on the wasserstein space projection method. Computers Mathematics with Applications, 2025. URL https://www.sciencedirect.com/science/article/abs/pii/ S001046552500368

Elie Khoury, Thomas Hain, et al. Automatic speech recognition for non-native english. arXiv preprint arXiv:2503.06924, 2025. URL https://arxiv.org/abs/2503.06924.